# A Model to Search for Synthesizable Molecules

**John Bradshaw**
University of Cambridge
MPI for Intelligent Systems
jab255@cam.ac.uk

**Brooks Paige**
University of Cambridge
The Alan Turing Institute
bpaige@turing.ac.uk

**Matt J. Kusner**
University College London
The Alan Turing Institute
m.kusner@ucl.ac.uk

**Marwin H. S. Segler**
BenevolentAI
Westfälische Wilhelms-Universität Münster
marwin.segler@benevolent.ai

**José Miguel Hernández-Lobato**
University of Cambridge
The Alan Turing Institute
Microsoft Research Cambridge
jmh233@cam.ac.uk

## Abstract

Deep generative models are able to suggest new organic molecules by generating strings, trees, and graphs representing their structure. While such models allow one to generate molecules with desirable properties, they give no guarantees that the molecules can actually be synthesized in practice. We propose a new molecule generation model, mirroring a more realistic real-world process, where (a) reactants are selected, and (b) combined to form more complex molecules. More specifically, our generative model proposes a bag of initial reactants (selected from a pool of commercially-available molecules) and uses a reaction model to predict how they react together to generate new molecules. We first show that the model can generate diverse, valid and unique molecules due to the useful inductive biases of modeling reactions. Furthermore, our model allows chemists to interrogate not only the properties of the generated molecules but also the feasibility of the synthesis routes. We conclude by using our model to solve retrosynthesis problems, predicting a set of reactants that can produce a target product.

## 1 Introduction

The ability of machine learning to generate structured objects has progressed dramatically in the last few years. One particularly successful example of this is the flurry of developments devoted to generating small molecules [14, 46, 27, 10, 49, 11, 20, 30, 55, 41, 2]. These models have been shown to be extremely effective at finding molecules with desirable properties: drug-like molecules [14], biological target activity molecules [46], and soluble molecules [11].

However, these improvements in molecule discovery come at a cost: these methods do not describe *how to synthesize such molecules*, a prerequisite for experimental testing. Traditionally, in computer-aided molecular design, this has been addressed by virtual screening [48], where molecule data sets $|D| \approx 10^8$, are first generated via the expensive combinatorial enumeration of molecular fragments stitched together using hand-crafted bonding rules, and then are scored in an $\mathcal{O}(|D|)$ step.

In this paper we propose a generative model for molecules (shown in Figure 1) that describes how to make such molecules from a set of commonly-available reactants. Our model first generates a set of reactant molecules, and second maps them to a predicted product molecule via a reaction prediction model. It allows one to simultaneously search for better molecules and describe how such molecules can be made. By closely mimicking the real-world process of designing new molecules, we show that our model: 1. Is able to generate a wide range of molecules not seen in the training data; 2.

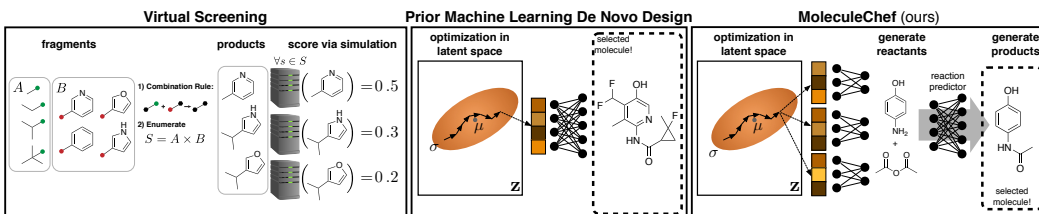

Figure 1: An overview of approaches used to find molecules with desirable properties. *Left*: Virtual screening [48] aims to find novel molecules by the (computationally expensive) enumeration over all possible combinations of fragments. *Center*: More recent ML approaches, eg [14], aim to find useful, novel molecules by optimizing in a continuous latent space; however, there are no clues to whether (and how) these molecules can be synthesized. *Right*: We approach the generation of molecules through a multistage process mirroring how complex molecules are created in practice, while maintaining a continuous latent space to use for optimization. Our model, MOLECULE CHEF, first finds suitable reactants which then react together to create a final molecule.

Addresses practical synthesis concerns such as reaction stability and toxicity; and 3. Allows us to propose new *reactants* for given target molecules that may be more practical to manage.

## 2   Background

We start with an overview of traditional computational techniques to discover novel molecules with desirable properties. We then review recent work in machine learning (ML) that seeks to improve parts of this process. We then identify aspects of molecule discovery we believe deserve much more attention from the ML community. We end by laying out our contributions to address these concerns.

### 2.1   Virtual Screening

To discover new molecules with certain properties, one popular technique is *virtual screening* (VS) [48, 17, 37, 8, 34]. VS works by (a) enumerating all combinations of a set of building-block molecules (which are combined via virtual chemical bonding rules), (b) for each molecule, calculating the desired properties via simulations or prediction models, (c) filtering the most interesting molecules to synthesize in the lab. While VS is general, it has the important downside that the generation process is not targeted: VS needs to get lucky to find molecules with desirable properties, it does not search for them. Given that the number of possible drug-like compounds is estimated to be $\in [10^{23}, 10^{100}]$ [52], the chemical space usually screened in VS $\in [10^{7}, 10^{10}]$ is tiny. Searching in combinatorial fragment spaces has been proposed, but is limited to simpler similarity queries [38].

### 2.2   The Molecular Search Problem

To address these downsides, one idea is to replace this full enumeration with a search algorithm; an idea called *de novo-design* (DND) [42]. Instead of generating a large set of molecules with small variations, DND searches for molecules with particular properties, recomputes them for the newfound molecules, and searches again. We call this **the molecular search problem**. Early work on the molecular search problem used genetic algorithms, ant-colony optimization, or other discrete search techniques to make local changes to molecules [16]. While more directed than library-generation, these approaches still explored locally, limiting the diversity of discovered molecules.

The first work to apply current ML techniques to this problem was Gómez-Bombarelli et al. [14] (in a late 2016 preprint). Their idea was to search by learning a mapping from molecular space to continuous space and back. With this mapping it is possible to leverage well-studied optimization techniques to do search: local search can be done via gradient descent and global search via Bayesian optimization [50, 13]. For such a mapping, the authors chose to represent molecules as SMILES strings [54] and leverage advances in generative models for text [5] to learn a character variational autoencoder (CVAE) [26]. Shortly after this work, in an early 2017 preprint, Segler et al. [46] trained recurrent neural networks (RNNs) to take properties as input and output SMILES strings with these properties, with molecular search done using reinforcement learning (RL).

**In Search of Molecular Validity.** However, the SMILES string representation is very brittle: if individual characters are changed or swapped, it may no longer represent any molecule (called an *invalid* molecule). Thus, the CVAE often produced invalid molecules (in one experiment, Kusner et al. [27] sampling from the continuous space, produced valid molecules only $0.7\%$ of the time). To address this validity problem, recent works have proposed using alternative molecular representations such as parse trees [27, 10] or graphs [49, 11, 29, 20, 30, 55, 21, 24, 41], where some of the more recent among these enforce or strongly encourage validity [20, 30, 55, 24, 41]. In parallel, there has been work based on RL that has aimed to learn a validity function during training directly [15, 18].

### 2.3 The Molecular Recipe Problem

Crucially, all of the works in the previous section solving **the molecular search problem** focus purely on optimizing molecules towards desirable properties. These works, in addressing the downsides of VS, removed a benefit of it: knowledge of the synthesis pathway of each molecule. Without this we do not know *how practical it is to make ML-generated molecules.*

To address this concern is to address **the molecular recipe problem**: what molecules are we able to make, given a set of readily-available starting molecules? So far, this problem has been addressed independently of the molecular search problem through synthesis planning (SP) [47]. SP works by recursively deconstructing a molecule. This deconstruction is done via (reversed) *reaction predictors*: models that predict how reactant molecules produce a product molecule. More recently, novel ML models have been designed for reaction prediction [53, 45, 19, 43, 6, 44].

### 2.4 This Work

In this paper, we propose to address both **the molecular search problem** and **the molecular recipe problem** jointly. To do so, we propose a generative model over molecules using the following map: First, a mapping from continuous space to a set of known, reliable, easy-to-obtain reactant molecules. Second a mapping from this set of reactant molecules to a final product molecule, based on a reaction prediction model [53, 45, 19, 44, 6]. Thus our generative model not only generates molecules, but also *a synthesis route using available reactants.* This addresses the molecular recipe problem, and also the molecular search problem, as the learned continuous space can also be used for search. Compared to previous work, in this work we are searching for new molecules through virtual chemical reactions, more directly simulating how molecules are actually discovered in the lab.

Concretely, we argue that our model, which we shall introduce in the next section, has several advantages over the current deep generative models of molecules reviewed previously:

**Better extrapolation properties** Generating molecules through graph editing operations, representing reactions, we hope gives us strong inductive biases for extrapolating well.

**Validity of generated molecules** Naive generation of molecular SMILES strings or graphs can lead to molecules that are invalid. Although the syntactic validity can be fixed by using masking [27, 30], the molecules generated can often still be semantically invalid. By generating molecules from chemically stable reactants by means of reactions, our model proposes more semantically valid molecules.

**Provide synthesis routes** Proposed molecules from other methods can often not be evaluated in practice, as chemists do not know how to synthesize them. As a byproduct of our model we suggest synthetic routes, which could have a useful, practical value.

## 3 Model

In this section we describe our model[1]. We define the set of all possible valid molecular graphs as $\mathcal{G}$, with an individual graph $g \in \mathcal{G}$ representing the atoms of a molecule as its nodes, and the type of bonds between these atoms (we consider single, double and triple bonds) as its edge types. The set of common reactant molecules, easily procurable by a chemist, which we want to act as building blocks for any final molecule is a subset of this, $\mathcal{R} \subset \mathcal{G}$.

As discussed in the previous section (and shown in Figure 1) our generative model for molecules consists of the composition of two parts: (1) a decoder from a continuous latent space, $\mathbf{z} \in \mathbb{R}^m$, to a bag (ie multiset[2]) of easily procurable reactants, $\mathbf{x} \subset \mathcal{R}$; (2) a reaction predictor model that transforms this bag of molecules into a multiset of product molecules $\mathbf{y} \subset \mathcal{G}$.

The benefit of this approach is that for step (2) we can pick from several existing reaction predictor models, including recently proposed methods that have used ML techniques [25, 45, 44, 6, 9]. In this work we use the Molecular Transformer (MT) of Schwaller et al. [44], as it has recently been shown to provide state-of-the-art performance in this task [44, Table 4].

This leaves us with the task of (1), learning a way to decode to (and encode from) a bag of reactants, using a parameterized encoder $q(\mathbf{z}|\mathbf{x})$ and decoder $p(\mathbf{x}|\mathbf{z})$. We call this co-occurrence model MOLECULE CHEF, and by moving around in the latent space we can *select* using MOLECULE CHEF different "bags of reactants".

Again there are several viable options of how to learn MOLECULE CHEF. For instance one could choose to use a VAE for this task [26, 40]. However, when paired with a complex decoder these models are often difficult to train [5, 1], such that much of the previous work for generating graphs has has tuned down the KL regularization term in these models [30, 27]. We therefore instead propose using the WAE objective [51], which involves minimizing

$$L = \mathbb{E}_{\mathbf{x} \sim \mathcal{D}} \mathbb{E}_{q(\mathbf{z}|\mathbf{x})} \left[ c(\mathbf{x}, p(\mathbf{x}|\mathbf{z})) \right] + \lambda D \left( \mathbb{E}_{\mathbf{x} \sim \mathcal{D}} \left[ q(\mathbf{z}|\mathbf{x}) \right], p(\mathbf{z}) \right)$$

where $c$ is a cost function, that enforces the reconstructed bag to be similar to the encoded one. $D$ is a divergence measure, which is weighted in relative importance by $\lambda$, that forces the marginalised distribution of all encodings to match the prior on the latent space. Following Tolstikhin et al. [51] we use the maximum mean discrepancy (MMD) divergence measure, with $\lambda = 10$ and a standard normal prior over the latents. We choose $c$ so that this first term matches the reconstruction term we would obtain in a VAE, i.e. with $c(\mathbf{x}, \mathbf{z}) = -\log p(\mathbf{x}|\mathbf{z})$. This means that the objective only differs from a VAE in the second, regularisation term, such that we are not trying to match each encoding to the prior but instead the marginalised distribution over all datapoints. Empirically, we find that this trains well and does not suffer from the same local optimum issues as the VAE.

### 3.1 Encoder and Decoder

We can now begin describing the structure of our encoder and decoder. In these functions it is often convenient to work with $n$-dimensional vector embeddings of graphs, $\mathbf{m}_g \in \mathbb{R}^n$. Again we are faced with a series of possible alternative ways to compute these embeddings. For instance, we could ignore the structure of the molecular graph and learn a distinct embedding for each molecule, or use fixed molecular fingerprints, such as Morgan Fingerprints [33]. We instead choose to use deep graph neural networks [32, 12, 3] that can produce graph-isomorphic representations.

Deep graph neural networks have been shown to perform well on a variety of tasks involving small organic molecules, and their advantages compared to the previously mentioned alternative approaches are that (1) they take the structure of the graph into account and (2) they can learn which characteristics are important when forming higher-level representations. In particular in this work we use 4 layer Gated Graph Neural Networks (GGNN) [28]. These compute higher-level representations for each node. These node-level representations in turn can be combined by a weighted sum, to form a graph-level representation invariant to the order of the nodes, in an operation referred to as an aggregation transformation [22, §3].

**Encoder** The structure of MOLECULE CHEF's encoder, $q(\mathbf{z}|\mathbf{x})$, is shown in Figure 2. For the $i$th data point the encoder has as input the multiset of reactants $\mathbf{x_i} = \{x_1^i, x_2^i, \cdots \}$. It first computes the representation of each individual reactant molecule graph using the GGNN, before summing these representations to get a representation that is invariant to the order of the multiset. A feed forward network is then used to parameterize the mean and variance of a Gaussian distribution over $\mathbf{z}$.

**Decoder** The decoder, $p(\mathbf{x}|\mathbf{z})$, (Figure 3) maps from the latent space to a multiset of reactant molecules. These reactants are typically small molecules, which means we could fit a deep generative

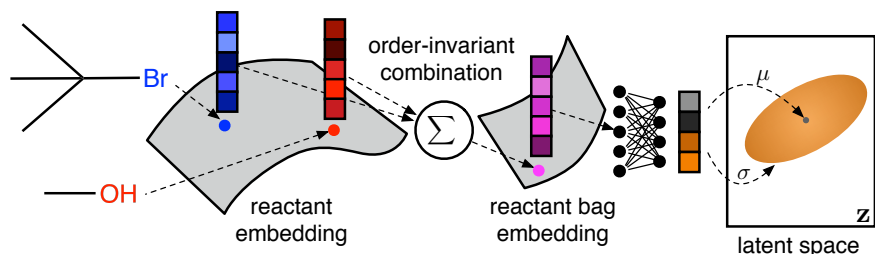

Figure 2: The encoder of MOLECULE CHEF. This maps from a multiset of reactants to a distribution over latent space. There are three main steps: (1) the reactants molecules are embedded into a continuous space by using GGNNs [28] to form molecule embeddings; (2) the molecule embeddings in the multiset are summed to form one order-invariant embedding for the whole multiset; (3) this is then used as input to a neural network which parameterizes a Gaussian distribution over $\mathbf{z}$.

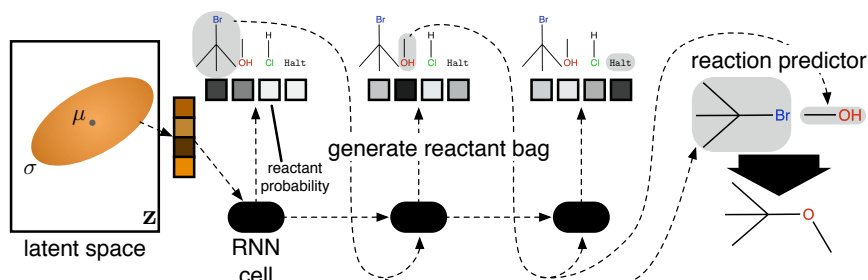

Figure 3: The decoder of MOLECULE CHEF. The decoder generates the multiset of reactants in sequence through calls to a RNN. At each step the model picks either one reactant from the pool or to halt, finishing the sequence. The latent vector, $\mathbf{z}$, is used to parameterize the initial hidden layer of the RNN. Reactants that are selected are fed back into the RNN on the next step. The reactant bag formed is later fed through a reaction predictor to form a final product.

model which produces them from scratch. However, to better mimic the process of selecting reactant molecules from an easily obtainable set, we instead restrict the output of the decoder to pick the molecules from a fixed set of reactant molecules, $\mathcal{R}$.

This happens in a sequential process using a recurrent neural network (RNN), with the full process described in Algorithm 1. The latent vector, $\mathbf{z}$ is used to parametrize the initial hidden layer of the RNN. The selected reactants are fed back in as inputs to the RNN at the next generation stage. Whilst training we randomly sample the ordering of the reactants, and use teacher forcing.

## 3.2 Adding a predictive penalty loss to the latent space

As discussed in section 2.2 we are interested in using and evaluating our model's performance in the **molecular search problem**, that is using the learnt latent space to find new molecules with desirable properties. In reality we would wish to measure some complex chemical property that can only be measured experimentally. However, as a surrogate for this, following [14], we optimize instead for the QED (Quantitative Estimate of Drug-likeness [4]) score of a molecule, $\mathbf{w}$, as a deterministic mapping from molecules to this score, $\mathbf{y} \mapsto \mathbf{w}$, exists in RDKit [39].

To this end, in a similar manner to Liu et al. [30, §4.3] & Jin et al. [20, §3.3], we can simultaneously train a 2 hidden layer property predictor NN for use in local optimization tasks. This network tries to predict the QED property, $\mathbf{w}$, of the final product $\mathbf{y}$ from the latent encoding of the associated bag of reactants. The use of this property predictor network for local optimization is described in Section 4.2.

**Algorithm 1** MOLECULE CHEF's Decoder
***

**Require:** $\mathbf{z}^i$ (latent space sample), GGNN (for embedding molecules), RNN (recurrent neural network), $\mathcal{R}$ (set of easy-to-obtain reactant molecules), $\mathbf{s}$ (learnt "halt" embedding), $\mathbf{A}$ (learnt matrix that projects the size of the latent space to the size of RNN's hidden space)

$\quad\mathbf{h}_0 \leftarrow \mathbf{A}\mathbf{z}^i$ ; $\mathbf{m}_0 \leftarrow \mathbf{0}$ {Start symbol}
$\quad$**for** $t = 1$ to $T_{\max}$ **do**
$\quad\quad\mathbf{h}_t \leftarrow \text{RNN}(\mathbf{m}_{t-1}, \mathbf{h}_{t-1})$ ; $\mathbf{B} \leftarrow \text{STACK}([\text{GGNN(g) for all g in } \mathcal{R}] + [\mathbf{s}])$
$\quad\quad\text{logits} \leftarrow \mathbf{h}_t \mathbf{B}^T$
$\quad\quad x_t \sim \text{softmax}(\text{logits})$
$\quad\quad$**if** $x_t = \text{HALT}$ **then**
$\quad\quad\quad$break {If the logit corresponding to the halt embedding is selected then we stop early}
$\quad\quad$**else**
$\quad\quad\quad\mathbf{m}_t \leftarrow \text{GGNN}(x_t)$
$\quad\quad$**end if**
$\quad$**end for**
$\quad$return $x_1, x_2, \cdots$
***

## 4 Evaluation

In this section we evaluate MOLECULE CHEF in (1) its ability to generate a diverse set of valid molecules; (2) how useful its learnt latent space is when optimizing product molecules for some property; and (3) whether by training a regressor back from product molecules to the latent space, MOLECULE CHEF can be used as part of a setup to perform retrosynthesis.

In order to train our model we need a dataset of reactant bags. For this we use the USPTO dataset [31], processed and cleaned up by Jin et al. [19]. We filter out reagents, molecules that form context under which the reaction occurs but do not contribute atoms to the final products, by following the approach of Schwaller et al. [43, §3.1].

We wish to use as possible reactant molecules only popular molecules that a chemist would have easy access to. To this end, we filter our training (using Jin et al. [19]'s split) dataset so that each reaction only contains reactants that occur at least 15 times across different reactions in the original larger training USPTO dataset. This leaves us with a dataset of 34426 unique reactant bags for training the MOLECULE CHEF. In total there are 4344 unique reactants. For training the baselines, we combine these 4344 unique reactants and the associated products from their different combinations, to form a training set for baselines, as even though MOLECULE CHEF has not seen the products during training, the reaction predictor has.

### 4.1 Generation

We begin by analyzing our model using the metrics favored by previous work[3] [20, 30, 29, 27]: validity, uniqueness and novelty. Validity is defined as requiring that at least one of the molecules in the bag of products can be parsed by RDKit. For a bag of products to be unique we require it to have at least one valid molecule that the model has not generated before in any of the previously seen bags. Finally, for computing novelty we require that the valid molecules not be present in the same training set we use for the baseline generative models.

In addition, we compute the Fréchet ChemNet Distance (FCD) [36] between the valid molecules generated by each method and our baseline training set. Finally in order to try to assess the *quality* of the molecules generated we record the (train-normalized) proportion of valid molecules that pass the quality filters proposed by Brown et al. [7, §3.3]; these filters aim to remove molecules that are *"potentially unstable, reactive, laborious to synthesize, or simply unpleasant to the eye of medicinal chemists"*.

***

[3]Note that we have extended the definition of these metrics to a bag (multiset) of products, given that our model can output multiple molecules for each reaction. However, when sampling 20000 times from the prior of our model, we generate single product bags 97% of the time, so that in practice most of the time we are using the same definition for these metrics as the previous work which always generated single molecules.

Table 1: Table showing the validity, uniqueness, novelty and normalized quality (all as %, higher better) of the products/or molecules generated from decoding from 20k random samples from the prior $p(\mathbf{z})$. Quality is the proportion of valid molecules that pass the quality filters proposed in Brown et al. [7, §3.3], normalized such that the score on the training set is 100. FCD is the Fréchet ChemNet Distance [36], capturing a notion of distance between the generated valid molecules and the training dataset (lower better). The uniqueness and novelty figures are also conditioned on validity. MT stands for the Molecular Transformer [44].

| Model Name | Validity | Uniqueness | Novelty | Quality | FCD |
|---|---|---|---|---|---|
| MOLECULE CHEF + MT | 99.05 | 95.95 | 89.11 | 95.30 | 0.73 |
| AAE [23, 35] | 85.86 | 98.54 | 93.37 | 94.89 | 1.12 |
| CGVAE [30] | 100.00 | 93.51 | 95.88 | 44.45 | 11.73 |
| CVAE [14] | 12.02 | 56.28 | 85.65 | 52.86 | 37.65 |
| GVAE [27] | 12.91 | 70.06 | 87.88 | 46.87 | 29.32 |
| LSTM [46] | 91.18 | 93.42 | 74.03 | 100.12 | 0.43 |

For the baselines we consider the character VAE (CVAE) [14], the grammar VAE (GVAE) [27], the AAE (adversarial autoencoder) [23], the constrained graph VAE (CGVAE) [30], and a stacked LSTM generator with no latent space [46]. Further details about the baselines can be found in the appendix.

The results are shown in Table 1. As MOLECULE CHEF decodes to a bag made up from a predefined set of molecules, those reactants going into the reaction predictor are valid. The validity of the final product is not 100%, as the reaction predictor can make non-valid edits to these molecules, but we see that in a high number of cases the products are valid too. Furthermore, what is very encouraging is that the molecules generated often pass the quality filters, giving evidence that the process of building molecules up by combining stable reactant building blocks often leads to stable products.

## 4.2 Local Optimization

As discussed in Section 3.2, when training MOLECULE CHEF we can simultaneously train a property predictor network, mapping from the latent space of MOLECULE CHEF to the QED score of the final product. In this section we look at using the gradient information obtainable from this network to do local optimization to find a molecule created from our reactant pool that has a high QED score.

We evaluate the local optimization of molecular properties by taking 250 bags of reactants, encoding them into the latent space of MOLECULE CHEF, and then repeatedly moving in the latent space using the gradient direction of the property predictor until we have decoded ten different reactant bags. As a comparison we consider instead moving in a random walk until we have also decoded to ten different reaction bags. In Figure 4 we look at the distribution of the best QED score found in considering these ten reactant bags, and how this compared to the QEDs started with.

When looking at individual optimization runs, we see that the QEDs vary a lot between different products even if made with similar reactants. However, Figure 4 shows that overall the distribution of the final best found QED scores is improved when purposefully optimizing for this. This is encouraging as it gives evidence of the utility of these models for the molecular search problem.

## 4.3 Retrosynthesis

A unique feature of our approach is that we learn a decoder from latent space to a bag of reactants. This gives us the ability to do retrosynthesis by training a model to map from products to their associated reactants' representation in latent space and using this in addition to MOLECULE CHEF's decoder to generate a bag of reactants. This process is highlighted in Figure 5. Although retrosynthesis is a difficult task, with often multiple possible ways to create the same product and with current state-of-the-art approaches built using large reaction databases and able to deal with multiple reactions [47], we believe that our model could open up new interesting and exciting approaches to this task. We therefore train a small network based on the same graph neural network structure used for MOLECULE CHEF followed by four fully connected layers to regress from products to latent space.

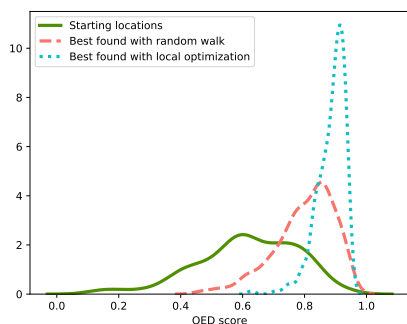

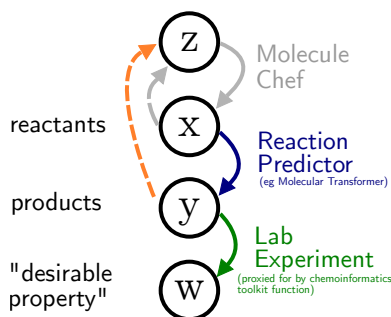

Figure 4: KDE plot showing that the distribution of the best QEDs found through local optimization, using our trained property predictor for QEDs, has higher mass over higher QED scores compared to the best found from a random walk. The starting locations' distribution (sampled from the training data) is shown in green. The final products, given a reactant bag are predicted using the MT [44].

Figure 5: Having learnt a latent space which can map to products through reactants, we can learn a regressor back from the suggested products to the latent space (orange dashed — — arrow shown) and couple this with MOLECULE CHEF's decoder to see if we can do retrosynthesis – the act of computing the reactants that create a particular product.

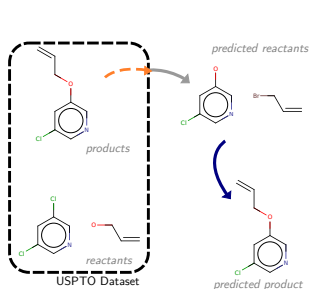

Figure 6: An example of performing retrosynthesis prediction using a trained regressor from products to latent space. This reactant-product pair has not been seen in the training set of MOLECULE CHEF. Further examples are shown in the appendix.

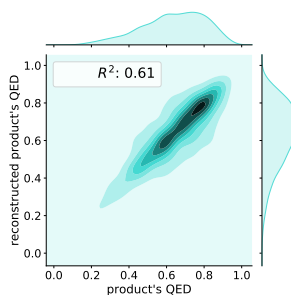

(a) Reachable Products

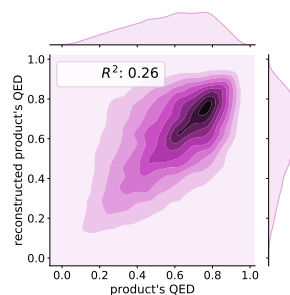

(b) Unreachable Products

Figure 8: Assessing the correlation between the QED scores for the original product and its reconstruction (see text for details). We assess on two portions of the test set, products that are made up of only reactants in MOLECULE CHEF's vocabulary are called 'Reachable Products', those that have at least one reactant that is absent are called 'Unreachable Products'.

A few examples of the predicted reactants corresponding to products from reactions in the USPTO test set, but which can be made in one step from the predefined possible reactants, are shown in Figure 6 and the appendix. We see that often this approach, although not always able to suggest the correct whole reactant bag, chooses similar reactants that on reaction produce similar structures to the original product we were trying to synthesize. While we would not expect this approach to retrosynthesis to be competitive with complex planning tools, we think this provides a promising new approach, which could be used to identify bags of reactants that produce molecules similar to a desired target molecule. In practice, it would be valuable to be pointed directly to molecules with similar properties to a target molecule if they are easier to make than the target, since it is the properties of the molecules, and not the actual molecules themselves, that we are after.

With this in mind, we assess our approach in the following way: (1) we take a product and perform retrosynthesis on it to produce a bag of reactants, (2) we transform this bag of reactants using the Molecular Transformer to produce a new *reconstructed* product, and then finally (3) we plot the resulting reconstructed product molecule's QED score against the QED score of the initial product. We evaluate on a filtered version of Jin et al. [19]'s test set split of USPTO, where we have filtered out any reactions which have the exact same reactant and product multisets as a reaction present in the set used to train Molecule Chef. In addition, we further split this filtered set into two sets: (i)

'Reachable Products', which are reactions in the test set that contain as reactants only molecules that are in MOLECULE CHEF's reactant vocabulary, and (ii) 'Unreachable Products', which have at least one reactant molecule that is not in the vocabulary.

The results are shown in Figure 8; overall we see that there is some correlation between the properties of products and the properties of their reconstructions. This is more prominent for the reachable products, which we believe is because our latent space is only trained on reachable product reactions and so is better able to model these reactions. Furthermore, some of the unreachable products may also require reactants that are not available in our pool of easily available reactants, at least when considering one-step reactions. However, given that unreachable products have at least one reactant which is not in Molecule Chef's vocabulary, we think it is very encouraging that there still is some, albeit smaller, correlation with the true QED. This is because it shows that our model can suggest molecules with similar properties made from reactants that are available.

### 4.4 Qualitative Quality of Samples

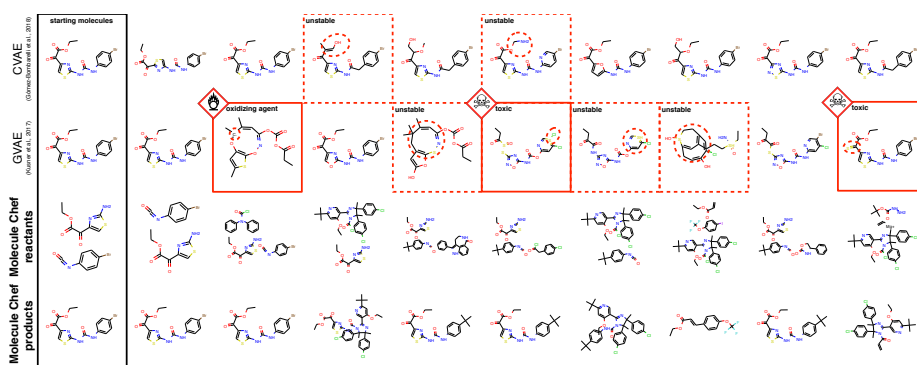

Figure 9: Random walk in latent space. See text for details.

In Figure 9 we show molecules generated from a random walk starting from the encoding of a particular molecule (shown in the left-most column). We compare the CVAE, GVAE, and MOLECULE CHEF (for MOLECULE CHEF we encode the reactant bag known to generate the same molecule). We showed all generated molecules to a domain expert and asked them to evaluate their properties in terms of their *stability*, *toxicity*, *oxidizing power*, *corrosiveness* (the rationales are provided in more detail in the Appendix). Many molecules produced by the CVAE and GVAE show undesirable features, unlike the molecules generated by MOLECULE CHEF.

## 5  Discussion

In this work we have introduced MOLECULE CHEF, a model that generates synthesizable molecules, by considering the products produced as a result of one-step reactions from a pool of pre-defined reactants. By constructing molecules through selecting reactants and running chemical reactions, while performing optimization in a continuous latent space, we can combine the strengths of previous VAE-based models and classical discrete de-novo design algorithms based on virtual reactions. As future work, we hope to explore how to extend our approach to deal with larger reactant vocabularies and multi-step reactions. This would allow the generation of a wider range of molecules, whilst maintaining our approach's advantages of being able to suggest synthetic routes and often producing *semantically valid* molecules.

**Acknowledgements**

This work was supported by The Alan Turing Institute under the EPSRC grant EP/N510129/1. JB also acknowledges support from an EPSRC studentship.

## Footnotes

[1]Further details can also be found in our appendix and code is available at `https://github.com/ john-bradshaw/molecule-chef`

[2]Note how we allow molecules to be present multiple times as reactants in our reaction, although practically many reactions only have one instance of a particular reactant.

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
