[Supplementary Material]

# Appendix for A Model to Search for Synthesizable Molecules

## A  Appendix

### A.1  Generation Benchmarks on ZINC

We also ran the baselines for the generation task on the ZINC dataset [Irwin et al., 2012]. The results are shown in Table 1.

Table 1: Table showing generation results for the baseline models when trained on ZINC dataset [Irwin et al., 2012]. The first four result columns show the validity, uniqueness, novelty and normalized quality (all as %, higher better) of the molecules generated from decoding from 20k random samples from the prior $p(\mathbf{z})$. Quality is the proportion of molecules that pass the quality filters proposed in Brown et al. [2019, §3.3], normalized such that the score on the USPTO derived training dataset (used in the main paper) is 100. FCD is the Fréchet ChemNet Distance [Preuer et al., 2018], capturing a notion of distance between the generated molecules and the USPTO derived training dataset used in the main paper.

| Model Name | Validity | Uniqueness | Novelty | Quality | FCD |
|---|---|---|---|---|---|
| AAE [Kadurin et al., 2017, Polykovskiy et al., 2018] | 87.64 | 100.00 | 99.99 | 96.12 | 7.27 |
| CGVAE [Liu et al., 2018] | 100.00 | 95.39 | 96.54 | 43.48 | 15.30 |
| CVAE [Gómez-Bombarelli et al., 2018] | 0.31 | 40.98 | 24.59 | 128.54 | 40.10 |
| GVAE [Kusner et al., 2017] | 3.66 | 85.23 | 95.08 | 38.86 | 27.31 |
| LSTM [Segler et al., 2017] | 95.71 | 99.98 | 99.93 | 108.68 | 7.99 |

## A.2 Further Random Walk Examples and Rationales for Expert Annotation

**Rationales for expert labels in Figure 9 in the main text.** Denoted using letters for letters for rows and numbers for columns. A3: Unstable, enol; A5: unstable, aminal; B2: reactive, radical; B4: unstable ring system; B5: toxic, reactive sulfur-chloride bond, unstable ring system; B6: unstable ring system; B7: unstable ring system; B9: toxic: thioketone.

Figure 1: Another example random walk in latent space. See §4.4 of main paper for further details. The rationals for the labels are (using letters for letters for rows and numbers for columns): A1: Unstable, gemthiolol; A7: unstable, gem-aminohydroxyl; B3: corrosive, acyl fluoride B5: corrosive, acyl fluoride; B6: corrosive, acyl chloride; B7: corrosive, acyl chloride; B9: corrosive, acyl bromide

Figure 2: Another example random walk in latent space. See §4.4 of main paper for further details. The rationals for the labels are (using letters for letters for rows and numbers for columns): A1: toxic, explosive, hydrazine, three consecutive aliphatic nitrogens; A2: toxic, hydrazine; A3: toxic, unstable, hydrazine, hemithioacetal; A4: toxic, hydrazine; A5: unstable, could be oxidized to 1,3,4-triazol, potentially also toxic due to N-N bond/hydrazine; A6: unstable, three-membered ring is antiaromatic; A7: toxic, hydrazine; A8: toxic, explosive, hydrazine, three consecutive aliphatic nitrogens. B4: unstable, hemiacetal; B5-B8: unstable, unfavorable ring systems

## A.3 Further Retrosynthesis Results

In this section we first provide more retrosynthesis examples before also describing an extra experiment in which we try to assess how well the retrosynthesis pipeline is at finding molecules with similar properties, even if not reconstructing the correct reactants themselves.

### A.3.1 Further Examples

Figure 3: Further examples of the predicted reactants associated with a given product for product molecules not in MOLECULE CHEF's training dataset, however with reactants belonging to MOLECULE CHEF's vocabulary (ie Reachable Dataset).

Figure 4: Further examples of the predicted reactants associated with a given product for product molecules not in MOLECULE CHEF's training dataset, with at least one reactants not part of MOLECULE CHEF's vocabulary (ie Unreachable Dataset).

## A.3.2 ChemNet Distances between Products and their Reconstructions

We also consider an experiment for which we analyze the Euclidean distance between the ChemNet embeddings of the product and the reconstructed product (found by feeding the original product through our retrosynthesis pipeline and then the Molecular Transformer). ChemNet embeddings are used when calculating the FCD score [Preuer et al., 2018] between molecule distributions, and so hopefully capture various properties of the molecule [Mayr et al., 2018]. Whilst learning MOLECULE CHEF we include a NN regressor from the latent space to the associated ChemNet embeddings, for which the MSE loss is minimized during training.

To try to establish an idea of how randomly chosen pairs of molecules in our dataset differ from each other, when measured using this metric, we provide a distribution of the distances of random pairs. This distribution is formed by taking each of the molecules in our dataset (consisting of all the reactants and their associated products) and matching it up with another randomly chosen molecule from this set, before measuring the Euclidean distance between the embeddings of each of these pairs.

The results are shown in Figure 5. We see that the distribution of distances between the products and their reconstructions has greater mass on smaller distances compared to the random pairs baseline.

(a) When evaluated on the portion of USPTO test set reactions for which both reactants are present in the MOLECULE CHEF's vocabulary.

(b) When evaluated on the portion of USPTO test set reactions for which at least one reactant is not present in the MOLECULE CHEF's vocabulary.

Figure 5: KDE plot showing the distribution of the Euclidean distances between the ChemNet embeddings [Preuer et al., 2018] of our product and reconstructed product.

## A.4 Details about our Dataset

In this section we provide further details about the molecules used in training our model and the baselines. We also describe details of the molecules used in the retrosynthesis experiments.

For MOLECULE CHEF's vocabulary we use reactants that occur at least 15 times in the USPTO train dataset, as processed and split by Jin et al. [2017]. This dataset uses reactions collected by Lowe [2012] from USPTO patents. In total we have 4344 reactants, and a training set of 34426 unique reactant bags for which these reactants co-occur. Each reactant bag is associated with a product.

For the baselines we train on these reactants and the associated products. This results in a dataset of approximately 37000 unique molecules, containing a wide variety of heavy elements: { 'Al', 'B', 'Br', 'C', 'Cl', 'Cr', 'Cu', 'F', 'I', 'K', 'Li', 'Mg', 'Mn', 'N', 'Na', 'O', 'P', 'S', 'Se', 'Si', 'Sn', 'Zn' } .

Some examples of the molecules found in the dataset are shown in Figure 6. Note that the large number of heavy atoms present, as well as the small overall dataset size, makes a challenging learning task compared to when using some of the more common benchmark datasets used elsewhere (such as ZINC [Irwin et al., 2012]).

We use examples from the USPTO test dataset when performing the retrosynthesis experiments. However, we first filter out any reactions for which the exact same reactant/product multisets tuple is

Figure 6: Examples of molecules found in the dataset we use for training the baselines. This is a subset of the molecules found in USPTO [Lowe, 2012]. It consists of the reactants that the MOLECULE CHEF can produce along with their corresponding products. It contains complex molecules with challenging structures to learn.

also present in the training data for MOLECULE CHEF[1]. Then we split the resultant dataset into two subsets. The first, which we refer to as the reachable dataset, contains only reactants in MOLECULE CHEF's vocabulary. The second, which we refer to as the unreachable dataset contains reactions with at least one reactant not in the vocabulary.

## A.5 Implementation Details

**Details of MOLECULE CHEF's architecture and parameters**

An overview of MOLECULE CHEF's architrecture can be seen in Figure 7. The encoder takes in a multiset of reactants and outputs the parameters of a Gaussian distribution over $z$. The decoder maps from the latent space to a multiset of reactant molecules. Both of these networks rely in turn on vector representations of molecules computed by a graph neural network. We provide details of these networks' architectures below as well as training details. Further information can be found in our code available at `https://github.com/john-bradshaw/molecule-chef`.

**Computing vector representations of molecular graphs using graph neural networks** For computing the vector representation of molecular graphs we used Gated Graph Neural Networks [Li et al., 2016], with the same network shared in both the encoder and decoder. We run these networks for 4 propagation steps and the node representations have a dimension of 101. We initialise the node reprsentations with the atom features shown in Table 2. The final step's node representation is projected down to a dimension of 50 by using a learnt linear projection. Graph level representations are formed from these node representations by performing a weighted sum.

Figure 7: Overview of MOLECULE CHEF showing how the encoder and decoder fit together.

Table 2: Atom features we use as input to the GGNN. These are calculated using RDKit.

| Feature | Description |
| --- | --- |
| Atom type | 72 possible elements in total, one hot |
| Degree | One hot (0, 1, 2, 3, 4, 5, 6, 7, 10) |
| Explicit Valence | One hot (0, 1, 2, 3, 4, 5, 6, 7, 8, 10, 12, 14) |
| Hybridization | One hot (SP, SP2, SP3, Other) |
| H count | integer |
| Electronegativity | float |
| Atomic number | integer |
| Part of an aromatic ring | boolean |

**Encoder**   The encoder sums the vector representations of the molecules present in the reactant multiset to get a 50 dimensional vector representation of the entire multiset. This representation is fed through a single hidden layer NN (with a hidden layer size of 200) to parameterise the mean and diagonal of the covariance matrix of a 25 dimensional multivariate-Gaussian distribution over $z$.

**Decoder**   The decoder maps from the the latent space, $z$, to a multiset of reactants. It does this through a sequential process, selecting one reactant at a time using a gated recurrent unit (GRU) [Cho et al., 2014] RNN. The parameters used for this GRU are shown in Table 3. The initial hidden state of the RNN is set using the result from a learnt linear projection of $z$. The final output of the GRU is fed through a single hidden layer NN (with a hidden size of 128) to form a final output vector. The dot product of this final output vector is formed with each of the possible reactant embeddings as well as the HALT embedding to form logits for the next output of the decoder. The embedding of the reactant selected is fed back in as input into the RNN at the next step.

Table 3: Parameters for GRU used in decoder

| Parameter | Value |
|---|---|
| GRU hidden size | 50 |
| GRU number of layers | 2 |
| GRU maximum number of steps | 5 |

**Property Predictor**    In §3.2 of the main paper we discuss how we also can train a property predictor from the latent space to a property of interest such as the QED, while traing the WAE. For the QED property predictor NN we use a fully connected network with two hidden layers, both with dimensionality of 40. The loss from this network is added to the WAE loss when training the model for the local optimization and retrosynthesis tasks.

**Training**    We train the WAE (and property predictor when applicable) for 100 epochs. We use the Adam optimizer [Kingma and Ba, 2015], with an initial learning rate of 0.001. We decay the learning rate by a factor of 10 every 40 epochs.

### Implementation Details for the Baselines in Section 4.1 of Main Paper

For the baselines in the generation section in the main paper we use the following implementations:

- CGVAE [Liu et al., 2018]: `https://github.com/microsoft/constrained-graph-variational-autoencoder`

- LSTM [Segler et al., 2017], : `https://github.com/BenevolentAI/guacamol_baselines`

- AAE [Kadurin et al., 2017, Polykovskiy et al., 2018]: `https://github.com/molecularsets/moses/tree/master/moses/aae`

- GVAE [Kusner et al., 2017]: `https://github.com/mkusner/grammarVAE`

- CVAE [Gómez-Bombarelli et al., 2018]: `https://github.com/mkusner/grammarVAE`

The LSTM baseline implementation follows Segler et al. [2017], which has as its alphabet a list of all individual element symbols, plus special characters used in SMILES strings. This differs from the alphabet used by the decoder in the Molecular Transformer [Schwaller et al., 2019], which instead extracts "bracketed" atoms directly from the training set; this means that a portion of a SMILES string such as `[OH+]` or `[NH2-]` would be represented as a single symbol, rather than as a sequence of five symbols. A regular expression can be used to extract a list of all such sequences from the training data. Effectively, this makes the trade off of increasing the alphabet size (from 47 to 203 items), while reducing the chance of making syntax errors or suggesting invalid charges. In practice we found very little qualitative or quantitative difference in the performance of the LSTM model for the two alphabets; for sake of consistency with MOLECULE CHEF we report the baseline using the larger alphabet.

For the CGVAE we decide to include element-charge-valence triplets that occur at least 10 times over all the molecules in the training data. At generation time we pick one starting node at random.

### Other Details

The majority of the experiments for MOLECULE CHEF were run on a NVIDIA Tesla K80 GPU. For running the Molecular Transformer and CGVAE, we used NVIDIA P100 and P40 GPUs, as the latter in particular required a large memory GPU for training on the larger datasets.

For MOLECULE CHEF we have not tried a wide range of hyperparameters. For the latent dimensionality we initially tried a dimension of 100 before trying and sticking with 25. Initially, we did not anneal the learning rate but found slightly improved performance by annealing it by a factor of 10 after 40 epochs. These changes were made after considering the reconstruction error of the model on the validation set (the validation dataset of USPTO restricted to the reactants in MOLECULE CHEF's vocabulary).

## Footnotes

[1]After canonicalisation and the removal of reagents, the USPTO train and test dataset has some reactions present in both sets.