[Reviews · NeurIPS 2019]

Reviewer 1



Originality: The proposed model (Molecule-Chef) is a novel combination of existing deep learning models - like Graph Neural Networks, RNNs, etc. to solve the task of molecule searching and to provide a synthesis recipe for the same. The authors also ensure the validity of products by restricting the latent space to chemical reactants that are readily available to chemists. Quality: The authors compare Molecule-Chef with state-of-the-art baselines and report improved/comparable results. By restricting the latent space, the model produces more valid molecular products as compared to other models. The paper also presents results of retrosynthesis, in which given some molecular products, the decoder of Molecule-Chef can be used to generate the possible combinations of reactants that were used to create the product. Clarity: The paper is well written and adequately covers the background for the task. More details could be added to the description of the model and tuning in the appendix. The authors could also include a figure showing the full architecture of Molecule-Chef since it has multiple parts. Significance: The paper has tried to tackle a useful problem of jointly solving molecule search and deciphering recipe tasks. It may be helpful for researchers developing methods to solve similar tasks.

Reviewer 2



The authors present an innovative model addressing two crucial problems in cheminformatics: the molecular search problem and the molecular recipe problem. The article is written a clear and understandable way, which leaves no loose ends by good argumentation and making nice use references. Furthermore, by providing the code makes the reviewing process easier. Based on this, the paper should be accepted as it is.

Reviewer 3



There are other previous methods that can generate valid molecules. For example, the relationship with the following paper is not clear. Hiroshi Kajino: "Molecular Hypergraph Grammar with Its Application to Molecular Optimization", ICML-19, Long Beach, CA, 2019. The authors in the above paper insist that most generated molecules are chemically valid. Did the authors compare with the method in the above paper? The detailed of the proposed method should be provided. I understand the page limitation. I would appreciate if the authors could provide the detailed explanations on the proposed method in the supplement. How did the authors use the method in the following paper? Marwin HS Segler, Mike Preuss, and Mark P Waller. Planning chemical syntheses with deep neural networks and symbolic AI. Nature, 555(7698):604, 2018. Data used in this paper were not provided, so the code did not work. There is no README file. Please provide data and the README file that describes how to run the program. Otherwise, it is impossible to evaluate the reproducibility of the proposed method. There are several typos. Line 98: we hope gives Line 128: has has Line 248: a target molecules

[Author Response · NeurIPS 2019]

We would like to thank the reviewers for their thoughtful comments. Detailed responses are below:

**Reviewer 1**  *[ ..on clarity.. ]* We appreciate your compliments on the writing and background section. We will follow your suggestion and add a figure showing the full Molecule Chef model as well as adding more details of the architecture and hyperparameters to the appendix (so that they can be found more conveniently than from looking through our code). Thank you for this suggestion.

*[ ..on another Fig. 6.. ]* Thanks, we will add to the appendix a third example figure, with a different initial molecule.

*[ ..on difference between correlations between reachable and unreachable products.. ]* We believe this difference is because our latent space is only trained on reachable product reactions and so is better able to model these reactions. Furthermore, some of the unreachable products may also require reactants that are not available in our pool of easily available reactants, at least when considering one-step reactions. Including larger reactant vocabularies and predicting of multi-stage reactions are two things we wish to look into in future work. However, given that unreachable products have at least one reactant which is not in Molecule Chef's vocabulary, we think it is very encouraging that there still is some, albeit smaller, correlation with the true QED. This is because it shows that our model can suggest molecules with similar properties made from reactants that are available. Thanks for bringing this up, we will clarify this in the paper.

*[ ..on rearranging some of the dataset details from the appendix.. ]* Thanks for the suggestion, we will do this.

**Reviewer 2**  Thank you for your positive comments and for saying that the 'paper should be accepted as it is'. We are happy to hear that you found the provided code useful. We also appreciate you highlighting the importance of the problem our model addresses and for recognizing our contributions, in particular the built-in generation of synthesis routes for suggested products and the generation of chemically sensible samples, 'in contrast with previous works showing undesirable molecular features'.

[..on your 'minor comment' regarding our section 4.4..] We obtained these annotations from an expert organic chemist. For example, in the first row of Figure 6, the 4th compound is unstable because it contains an enol, which will tautomerize (transform itself) to an aldehyde quickly. The 6th molecule in this row contains an aminal, which will rapidly decompose into the free acid, formaldehyde and ammonia. We will have them make their annotation criteria explicit in the paper. Thanks for this suggestion.

**Reviewer 3**  *[ ..on comparing to the Kajino ICML 2019 paper.. ]* We think this is missing the point of our work. We are happy to cite this paper, but ultimately our goal is not to compare with every last molecular generative model (there are currently more than 15). Instead we wanted a comprehensive comparison with strong baseline methods (LSTM, CVAE, GVAE) as well as state-of-the-art methods (AAE, CGVAE). In our paper we argued that our method has two key advantages over these previous methods, which also holds over the one you cite.

First, it ensures that the generated molecules *can be synthesized* (and provides these synthetic routes), while remaining competitive with current methods that do not have this restriction. To our knowledge, our model is the first to ensure synthesizability whilst jointly optimizing. Suggesting molecules that can actually be made is key in practical drug and materials design. Second, many of the previous models often generate chemically unstable molecules, which is not captured in validity measures. The CGVAE that we compare against, as well as the MHG-VAE from the paper you cite, are guaranteed to generate 100% valid molecules. However, this just ensures that the molecules can be parsed by the chemoinformatics library RDKit, e.g. by ensuring correct valencies. These models do not guarantee that the suggested molecules are *chemically sensible* (ie are stable enough to exist on earth) or are non-toxic. Whereas we show that the Molecule Chef is able to generate realistic, sensible and safe molecules (see Figure 6). Additional evidence of this is the 'Quality' GuacaMol score [6] in Table 1 which gives low scores to molecules that are "potentially unstable, reactive, laborious to synthesize, or simply unpleasant to the eye of medicinal chemists", and also in Appendix A.2.

*[ ..on providing further details of our method in the supplement.. ]* We will add further details of our architecture to the supplement.

*[ ..on the Segler et al.'s method.. ]* We believe there is some confusion here: we did not use the above method. The Segler model is only able to provide a plan towards a given molecule. It cannot perform any optimization of molecules.

*[ ..on adding a detailed readme and datasets to the provided code.. ]* Our plan was always to open-source the code on publication with a detailed README, links to the datasets/weights, and a Docker Image (defining our package environment) to make it easy to immediately use our code. The data was too large to upload at submission time, but we uploaded the code to get feedback on its organization, and for additional clarification on architecture/hyperparameters.

[Meta-Review · NeurIPS 2019]

The authors propose a deep learning method to generate molecular structures that are synthesizable and give a recipe for their synthesis from reactant molecules. The reviewers fond the contribution be significant for practioners. The authors addresed the unclear points in their response.